# Postnatal checks and primary care consultations in the year following childbirth: an observational cohort study of 309 573 women in the UK, 2006–2016

Holly Christina Smith [ID],[1] Sonia Saxena [ID],[2] Irene Petersen [ID] [1]

[1]Department of Primary Care and Population Health, University College London, London, UK
[2]School of Public Health, Imperial College London, London, UK

**Correspondence to**
Holly Christina Smith;
holly.dorning.18@ucl.ac.uk

## ABSTRACT

**Objective** To describe women's uptake of postnatal checks and primary care consultations in the year following childbirth.

**Design** Observational cohort study using electronic health records.

**Setting** UK primary care.

**Participants** Women aged 16–49 years who had given birth to a single live infant recorded in The Health Improvement Network (THIN) primary care database in 2006–2016.

**Main outcome measures** Postnatal checks and direct consultations in the year following childbirth.

**Results** We examined 1 427 710 consultations in 309 573 women who gave birth to 241 662 children in 2006–2016. Of these women, 78.7% (243 516) had a consultation at the time of the postnatal check, but only 56.2% (174 061) had a structured postnatal check documented. Teenage women (aged 16–19 years) were 12% less likely to have a postnatal check compared with those aged 30–35 years (incidence rate ratio (IRR) 0.88, 95% CI 0.85 to 0.91) and those living in the most deprived versus least deprived areas were 10% less likely (IRR 0.90, 95% CI 0.88 to 0.92). Women consulted on average 4.8 times per woman per year and 293 049 women (94.7%) had at least one direct consultation in the year after childbirth. Consultation rates were higher for those with a caesarean delivery (7.7 per woman per year, 95% CI 7.7 to 7.8). Consultation rates peaked during weeks 5–10 following birth (11.8 consultations/100 women) coinciding with the postnatal check.

**Conclusions** Two in 10 women did not have a consultation at the time of the postnatal check and four in 10 women have no record of receiving a structured postnatal check within the first 10 weeks after giving birth. Teenagers and those from the most deprived areas are among the least likely to have a check. We estimate up to 350 400 women per year in the UK may be missing these opportunities for timely health promotion and to have important health needs identified following childbirth.

## INTRODUCTION

Providing high-quality comprehensive postnatal primary care is a global goal for improving maternal and child health in the first year of life.[1] In the United Kingdom (UK), nearly 800 000 women give birth each year[2] and every woman has access to midwives and health visitors for the first few days after delivery. They are then discharged to the care of their general practitioner (GP) who invites them for a planned postnatal check 6–8 weeks after birth,[3] as recommended by The National Institute for Health and Care Excellence (NICE)[4] and WHO as part of routine postnatal care.[5] The maternal postnatal check provides a unique and timely opportunity for new mothers and healthcare professionals to evaluate their physical and mental health and assess how women are recovering after pregnancy and birth.[6] The postnatal check is also a point where women and primary care health professionals can discuss breastfeeding, postpartum contraception, smoking cessation, return to physical activity, and dietary advice, particularly after gestational weight gain. GPs also play a role in supporting parents to cope with managing day-to-day care and minor illness of infants, and identifying safeguarding concerns for new mothers and their babies.

The postnatal period is typically defined as the first 6 weeks after childbirth[7] and previous

studies estimate between 47% and 83% of women will report at least one health problem around 8 weeks postpartum.[8–10] Historically, it was anticipated women would recover from pregnancy within this time[11]; however, there is increasing evidence that women have ongoing health needs throughout the first year[12] and even longer.[13 14] For example, up to 5% of women will require ongoing management of medical complications of gestational diabetes[15]; and others may need support with postpartum conditions such as postnatal depression which occurs in one in six women.[16] Previous research may underestimate maternal morbidity as many women do not report symptoms or may be reluctant to seek support after childbirth.[17]

## Strengths and limitations of this study

► A major strength of this study is that it is among the largest of the population-based studies to date of postnatal care in the first 12 months.

► We drew on data from electronic health records which reflect real-world clinical practice in UK primary care.

► We are limited by what is recorded in a woman's electronic health record and genuine postnatal checks may have occurred in reality which were not accurately coded in primary care data.

► This study may underestimate the number of women who had a postnatal check compared with estimates based on survey data.

From previous cross-sectional surveys from 1995 and 2010–2014, it would appear that up to 90% of women attend their postnatal checks, but selection bias of survey participants may overestimate this figure.[18–20] We found no contemporary studies showing patterns of primary care use for women following childbirth. The aim of this observational cohort study was to examine the prevalence of postnatal checks, explore factors associated with having a postnatal check and primary care consultations for women in the first year after childbirth.

## METHODS
### UK healthcare
In the UK, healthcare is free at the point of delivery for all residents as part of the National Health Service (NHS). Primary care is typically the first point of contact and is largely delivered by GPs and other healthcare professionals (nurses and health visitors) within a practice. Information about patients and their health are collected during primary care consultations. This information is primarily used for clinical care but is also widely used for research through large healthcare databases.

### Data source
We used one of the largest UK primary care databases, The Health Improvement Network (THIN) database. As of December 2016, THIN contained anonymised electronic health records for 16 million registered patients from 730 practices across the UK.[21] The database contains patient-level information on demographics, prescribing, symptoms, procedures, prevention, lifestyle factors and diagnostics. Consultations can be linked to give a comprehensive picture of someone's care. Diagnostic and symptomatic information is categorised using Read codes, a hierarchical coding system.[22] Additional Health Data codes classify prevention and lifestyle information. Socioeconomic information is captured through Townsend score which provides a measure of material deprivation based on where a person lives, unemployment, car ownership, home ownership and household overcrowding.[23] THIN is broadly representative of the UK population in terms of demographics, chronic disease and mortality; however, there is an over-representation of more affluent people.[24]

### Study population
A cohort of women who have given birth to a single live infant has previously been identified within THIN.[25] Births and date of childbirth were identified using a combination of an antenatal record, delivery record, postnatal care record, date of last menstrual period, or birth of a child matched to the mother's record. This pre-existing cohort of women excluded more complex births, including multiple deliveries (twins, triplets etc.) and those with a known miscarriage, termination or stillbirth, so it is not possible to include these women in our study. This cohort contains information on approximately 650 000 pregnancies/childbirths in 1990–2016. In this study, we included women aged 15–49 years who gave birth between 1 January 2006 and 31 December 2015 from this cohort. Data quality criteria were also applied whereby practices which did not have acceptable computer use (ACU) or acceptable mortality rates (AMR) by the date of childbirth were excluded. ACU is the date a practice was continuously entering on average at least two therapy records, one medical record and one additional health data record per patient per year[26]; and AMR is the date a practice has comparable mortality rates to the rest of the UK, given the size and demographics of the practice.[27] Women who had been registered at a practice for less than 6 months were also excluded. It was possible for women to have multiple childbirths in this study. Women were followed up for 12 months to identify their primary use after each childbirth, censoring for maternal death or practice transfer.

### Definition of variables
#### Postnatal check
We identified a postnatal check as any consultation at the time of the check (typically weeks 6–8) which had a specific Read code (beginning with '62R' or '62S') or Additional Health Data code ('1044100000' or '1044000000') identifying it as a postnatal visit and/or check. Some women may receive this check slightly earlier or later, and on reviewing the data this window was expanded to weeks 5–10 after birth to include a peak in consultations. We identified substantial variation in the use of these codes by practices and change over time (data not shown). Therefore, we used a second and more sensitive approach where we assumed any consultation at the time of the routine appointment (weeks 5–10) was an opportunity for a postnatal check. The results of this second approach are included in online supplemental materials.

#### Consultations
A primary care consultation was defined as any direct contact between a patient and a healthcare professional taking place: in practice, in a patient's home or by telephone. It was assumed only one consultation took place each day for each woman, therefore multiple records on the same date were grouped.

## Patient and childbirth characteristics

We stratified our analysis by maternal age, number of births, Townsend score (described previously), smoking status, year (2 year bands) and mode of delivery. Women were assigned to a 5 year band according to their age. We used information in the mother's records as well as children registered within the same household at the time of birth to assign number of births (categorised as: first, second, third or higher, or unknown). We used Townsend score fifths whereby each woman is assigned to one of five groups of deprivation, from least to most deprived. We assigned women's smoking status as 'current smoker' (record of smoking at any time in the year after childbirth), 'past smoker' (record of smoking or being a former smoker in the 2 years prior to childbirth and not a current smoker), 'non-smoker' or 'unknown'. Mode of delivery was determined using the identifying pregnancy/childbirth Read codes and was broadly grouped into 'caesarean', 'vaginal' and 'unknown' based on classifications developed previously.[25]

## Statistical analysis

A table was derived to show characteristics of women at each childbirth. The crude consultation rate was calculated as the total number of consultations per total person-years, stratified by characteristics. To explore variation across the first year, consultation rate was calculated as the number of consultations on each day with number of women registered with a practice on that day as the denominator. Women who died or transferred practice were censored from the denominator each day. To examine those who had a postnatal check, first we calculated the crude proportion of women with the outcome in each patient strata. To explore variation by characteristic in more detail, we examined the likelihood of having a postnatal check between 5 and 10 weeks in women with at least 5 weeks of follow-up and complete deprivation (Townsend score) information. We developed mixed-effects Poisson models to estimate how the likelihood of having a postnatal check between weeks 5 and 10 varied by maternal age, Townsend score, mode of delivery, number of births, smoking status and year. Three models were developed: unadjusted, age adjusted and age deprivation adjusted. Practice and woman (as women can have multiple childbirths) were included as random effects terms, and the log of follow-up time (between weeks 5 and 10) was included as an offset. All analyses were conducted using Stata V.16 (StataCorp, College Station, Texas, USA).

## Patient and public involvement

Two public panels interested in primary care research provided feedback on the study outline. The groups were supportive of the idea to identify attendance of postnatal checks; they suggested we clearly identify groups who do not attend and to examine differences in a woman's first versus subsequent childbirths. As a result, we have

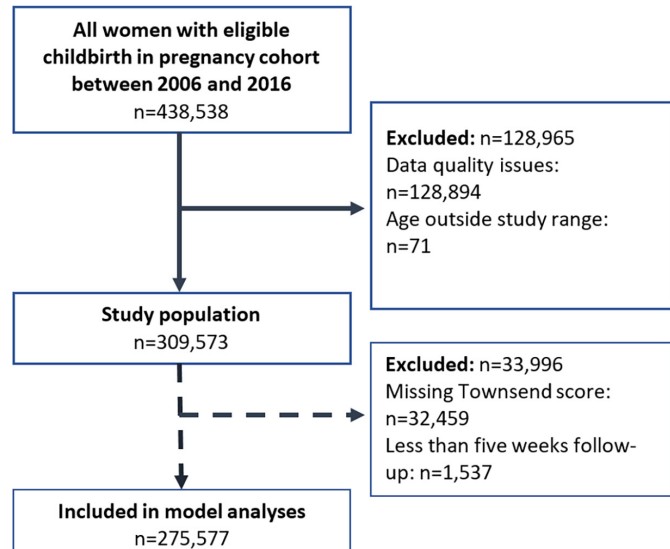

**Figure 1** Flow diagram showing application of study inclusion and exclusion criteria.

included additional analysis exploring attendance by patient characteristics and number of births.

## RESULTS

### Participants

Between 1 January 2006 and 31 December 2016, 438 538 pregnancies/childbirths were identified in the pregnancy cohort within THIN data. Study inclusion and exclusion criteria were applied to these records which resulted in a final sample of 309 573 childbirths (figure 1).

### Characteristics of women

We identified 309 573 childbirths in this study related to 241 662 women. At childbirth, a third of the women were aged 30–34 years (31.7%). There were 21.1% in the least deprived Townsend fifth compared with 16.0% in the most deprived (which is similar to the overall distribution in THIN).[24] Three quarters of women had a vaginal delivery and the rest had a caesarean birth (76.3% vs 23.7%). Of these, nearly half were a first birth (48%) and 22% were a second birth. Half of women were non-smokers (46.3%), compared with 11.2% being current smokers (table 1).

### Postnatal check

Overall, just over half of the women in our study (56%) had a structured postnatal check; 44% had no such records (table 1). In this crude analysis, younger women and those from the most deprived areas were less likely to have a postnatal check (48.1% of those aged 15–19 years vs 59.5% of those aged 35–39 years; and 47.7% of those from the most deprived area vs 62.7% from the least).

After excluding those with less than 5 weeks of follow-up information and missing deprivation information, 275 577 women were included in an additional analysis (figure 1). Those aged 15–19 years were 12% less likely (incidence rate ratio (IRR) 0.88, 95% CI 0.85 to 0.91) to have a postnatal check between weeks 5 and 10 relative

**Table 1** Characteristics of women who have given birth to a single live infant and proportion of those women with a structured postnatal check 5–10 weeks after childbirth

| Characteristic | All women (n) | Record of postnatal check in weeks 5–10 (n (% across the row)) |
|---|---|---|
| Overall | 309 573 | 174 061 (56.2) |
| Maternal age (years) | | |
| 15–19 | 9568 | 4599 (48.1) |
| 20–24 | 43 116 | 21 763 (50.5) |
| 25–29 | 77 698 | 42 417 (54.6) |
| 30–34 | 98 269 | 57 308 (58.3) |
| 35–39 | 64 171 | 38 154 (59.5) |
| 40–44 | 15 908 | 9347 (58.8) |
| 45–49 | 843 | 473 (56.1) |
| Townsend score | | |
| 1—least deprived | 58 583 | 36 752 (62.7) |
| 2 | 53 656 | 32 326 (60.3) |
| 3 | 62 023 | 35 413 (57.1) |
| 4 | 58 506 | 31 601 (54.0) |
| 5—most deprived | 44 346 | 21 138 (47.7) |
| Missing | 32 459 | 16 831 (51.9) |
| Mode of delivery | | |
| Vaginal delivery | 75 506 | 46 634 (61.8) |
| Caesarean | 23 426 | 14 384 (61.4) |
| Unknown | 210 641 | 113 043 (53.7) |
| No of births | | |
| First | 149 639 | 84 010 (56.1) |
| Second | 69 355 | 39 269 (56.6) |
| Third or higher | 20 113 | 10 781 (53.6) |
| Unknown | 70 466 | 40 001 (56.8) |
| Smoking status | | |
| Current smoker | 34 634 | 18 199 (52.6) |
| Former smoker | 85 592 | 47 464 (55.5) |
| Non-smoker | 143 349 | 82 420 (57.5) |
| Unknown | 45 998 | 25 978 (56.5) |
| Year group | | |
| 2006–2007 | 63 793 | 36 863 (57.8) |
| 2008–2009 | 66 319 | 38 124 (57.5) |
| 2010–2011 | 66 478 | 37 897 (57.0) |
| 2012–2013 | 63 180 | 34 896 (55.2) |
| 2014–2015 | 49 803 | 26 281 (52.8) |

to women aged 30–35 years (table 2). Similarly, women from the most deprived areas were 10% less likely (IRR 0.90, 95% CI 0.88 to 0.92) to have a postnatal check relative to those from the least deprived areas. Differences across other characteristics were less pronounced; and, adjusting for age, and age and deprivation, had little impact on differences across these other characteristics. The same trend across age and deprivation was identified when using a more sensitive approach to identify a potential postnatal check (any consultation between weeks 5 and 10), but with a higher proportion of women (78.7%) having a consultation (see online supplemental material).

A small proportion of women in our study (18 723, 6.0%) had a consultation in the first 4 weeks but did not have a consultation in weeks 5–10 (see online supplemental material). This compares to a much greater proportion of women (89 605, 28.9%) who had both an early consultation and one in weeks 5–10.

## Primary care consultations

Following the 309 573 childbirths, the majority (94.7%, n=293 049) of women had at least one direct consultation in the year after childbirth. A total of 1 427 710 direct consultations were identified, with women consulting on average 4.8 times/person-year (table 3). The largest difference in consultation rate compared with the average is seen in those who had a caesarean delivery (7.7/person-year, 95% CI 7.7 to 7.8) and in current smokers (5.9/person-year, 95% CI 5.9 to 5.9). Consultation rate decreased over time, from 4.9/person-year (95% CI 4.9 to 4.9) in 2006–2007 to 4.2/person year (4.2–4.3) in 2014–2015. Consultation rates were broadly similar across other characteristics.

Across the first year the consultation rate was highest between weeks 5 and 10, with a peak of 11.8 consultations/100 women in week 6, coinciding with the postnatal check. Following this, the consultation rate fell to an average of 1 consultation/100 women (figure 2).

## DISCUSSION
### Main findings

We found that eight in 10 women had a consultation at the time of the postnatal check; however, only half of women had a record of receiving a structured postnatal check. Teenage women (aged 15–19 years) were 12% less likely to have a postnatal check compared with older women aged 30–35 years, and those living in the most deprived areas were 10% less likely compared with women from least deprived areas. Women consulted on average 4.8 times per year in the year after childbirth and 94.7% of women had at least one consultation in the year after childbirth. Those who had a caesarean delivery and smokers had higher than average consultation rates (7.7 times per woman per year and 5.9 times per woman per year respectively). Consultation rates decreased over time (from 4.9 times per woman per year in 2006–2007 to 4.2 times in 2014–2015). Across the first year, the consultation rate is highest in week six with a peak of 11.8 consultations/100 women, which coincides with the postnatal check; after week 10 the consultation rate is flat with one consultation/100 women on each day.

### Study strengths and weaknesses

This is among the largest (299 688 person years) population-based studies to date of postnatal care in the

**Table 2** Mixed-effects Poisson estimates of the likelihood of having a postnatal check for women who had given birth to a single live infant by age, Townsend score, mode of delivery, number of births, smoking status and year group; unadjusted, and adjusted for age and deprivation

| Characteristic | n (%) | Record of postnatal check in weeks 5–10 | |
| --- | --- | --- | --- |
| | | Unadjusted (IRR (95% CI)) | Age and deprivation adjusted (IRR (95% CI)) |
| Overall | 275 577 | | |
| Maternal age (years) | | | |
| 15–19 | 8704 (3.2) | 0.88 (0.85 to 0.91) | 0.89 (0.87 to 0.92) |
| 20–24 | 38 503 (14.0) | 0.92 (0.91 to 0.94) | 0.93 (0.92 to 0.95) |
| 25–29 | 68 751 (25.0) | 0.97 (0.96 to 0.98) | 0.97 (0.96 to 0.99) |
| 30–34 | 86 889 (31.5) | 1 | 1 |
| 35–39 | 57 533 (20.9) | 1.00 (0.99 to 1.02) | 1.00 (0.99 to 1.01) |
| 40–44 | 14 428 (5.2) | 0.99 (0.97 to 1.01) | 0.99 (0.97 to 1.01) |
| 45–49 | 769 (0.3) | 0.96 (0.87 to 1.05) | 0.96 (0.87 to 1.05) |
| Townsend score | | | |
| 1—least deprived | 58 304 (21.2) | 1 | 1 |
| 2 | 53 370 (19.4) | 0.99 (0.98 to 1.01) | 1.00 (0.98 to 1.01) |
| 3 | 61 681 (22.4) | 0.97 (0.95 to 0.98) | 0.97 (0.96 to 0.99) |
| 4 | 58 165 (21.1) | 0.95 (0.93 to 0.97) | 0.96 (0.95 to 0.98) |
| 5—most deprived | 44 057 (16.0) | 0.90 (0.88 to 0.92) | 0.92 (0.90 to 0.93) |
| Mode of delivery | | | |
| Vaginal delivery | 68 202 (76.6) | 1 | 1 |
| Caesarean | 20 828 (23.4) | 1.02 (1.00 to 1.04) | 1.01 (0.99 to 1.03) |
| Unknown | 186 547 | 0.99 (0.97 to 1.01) | 0.99 (0.97 to 1.01) |
| No of births | | | |
| First | 132 164 (48.0) | 1 | 1 |
| Second | 62 535 (22.7) | 0.99 (0.98 to 1.01) | 0.98 (0.97 to 0.99) |
| Third or higher | 18 504 (6.7) | 0.95 (0.93 to 0.97) | 0.94 (0.92 to 0.96) |
| Unknown | 62 374 (22.6) | 0.96 (0.95 to 0.97) | 0.95 (0.94 to 0.96) |
| Smoking status | | | |
| Current smoker | 31 494 (11.4) | 0.94 (0.92 to 0.95) | 0.96 (0.95 to 0.98) |
| Former smoker | 76 941 (27.9) | 0.96 (0.95 to 0.97) | 0.96 (0.95 to 0.98) |
| Non-smoker | 126 497 (45.9) | 1 | 1 |
| Unknown | 40 645 (14.8) | 0.95 (0.94 to 0.97) | 0.95 (0.93 to 0.96) |
| Year group | | | |
| 2006–2007 | 58 606 (21.3) | 1 | 1 |
| 2008–2009 | 60 212 (21.9) | 0.99 (0.98 to 1.01) | 0.99 (0.98 to 1.01) |
| 2010–2011 | 59 183 (21.5) | 1.00 (0.99 to 1.02) | 1.00 (0.99 to 1.02) |
| 2012–2013 | 55 099 (20.0) | 0.99 (0.97 to 1.00) | 0.99 (0.97 to 1.01) |
| 2014–2015 | 42 477 (15.4) | 0.95 (0.94 to 0.97) | 0.95 (0.94 to 0.97) |

Practice and woman are included as random effect terms in all models.
IRR, incedence rate ratio.

first 12 months. The use of electronic health records provides a reflection of real-world clinical practice, allowing us to explore use of the postnatal check in a broad population (not reliant on patient participation). As with all studies of electronic health records, however, we are limited by what has been recorded in a woman's record. Which could mean patient, birth and consultation characteristics may be missing or not accurate. We also recognise the limitations in using Read codes/AHD codes only to identify a postnatal check as there

**Table 3** Crude consultation rate in the first year after childbirth of women who had given birth to a single live infant/person-year, by characteristic

| Characteristic | No of consultations | Person-years | Rate of consultations per person-year (95% CI) |
|---|---|---|---|
| Overall | 1 427 710 | 299 688 | 4.8 (4.8 to 4.8) |
| Maternal age (years) | | | |
| 15–19 | 46 087 | 9135 | 5.0 (5.0 to 5.1) |
| 20–24 | 211 905 | 41 212 | 5.1 (5.1 to 5.2) |
| 25–29 | 371 051 | 74 994 | 4.9 (4.9 to 5.0) |
| 30–34 | 437 873 | 95 385 | 4.6 (4.6 to 4.6) |
| 35–39 | 283 912 | 62 581 | 4.5 (4.5 to 4.6) |
| 40–44 | 72 941 | 15 561 | 4.7 (4.7 to 4.7) |
| 45–49 | 3941 | 821 | 4.8 (4.7 to 5.0) |
| Townsend score | | | |
| 1—least deprived | 269 502 | 57 128 | 4.7 (4.7 to 4.7) |
| 2 | 242 798 | 52 137 | 4.6 (4.6 to 4.7) |
| 3 | 285 632 | 59 972 | 4.8 (4.7 to 4.8) |
| 4 | 274 005 | 56 420 | 4.9 (4.8 to 4.9) |
| 5—most deprived | 212 965 | 42 609 | 5.0 (5.0 to 5.0) |
| Missing | 142 808 | 31 422 | 4.5 (4.5 to 4.6) |
| Mode of delivery | | | |
| Vaginal delivery | 511 769 | 73 251 | 7.0 (7.0 to 7.0) |
| Caesarean | 176 199 | 22 743 | 7.7 (7.7 to 7.8) |
| Unknown | 739 742 | 203 694 | 3.6 (3.6 to 3.6) |
| No of births | | | |
| First | 655 616 | 144 425 | 4.5 (4.5 to 4.6) |
| Second | 304 482 | 67 373 | 4.5 (4.5 to 4.5) |
| Third or higher | 100 156 | 19 722 | 5.0 (5.0 to 5.1) |
| Unknown | 367 456 | 68 168 | 5.4 (5.4 to 5.4) |
| Smoking status | | | |
| Current smoker | 201 192 | 34 045 | 5.9 (5.9 to 5.9) |
| Former smoker | 403 203 | 82 292 | 4.9 (4.9 to 4.9) |
| Non-smoker | 656 620 | 139 161 | 4.7 (4.7 to 4.7) |
| Unknown | 166 695 | 44 190 | 3.8 (3.8 to 3.8) |
| Year group | | | |
| 2006–2007 | 302 645 | 61 803 | 4.9 (4.9 to 4.9) |
| 2008–2009 | 318 827 | 64 356 | 5.0 (4.9 to 5.0) |
| 2010–2011 | 315 752 | 64 396 | 4.9 (4.9 to 4.9) |
| 2012–2013 | 285 875 | 60 974 | 4.7 (4.7 to 4.7) |
| 2014–2015 | 204 611 | 48 159 | 4.2 (4.2 to 4.3) |

is variation in the use of these by general practice and genuine checks may not be coded as such in primary care data. To account for this, we repeated our analysis using a more sensitive definition of a postnatal check (any consultation in week 5–10). While we found a larger proportion (78.7%) having been in contact with primary care, the overall trends in terms of the sociodemographic information were consistent between the two approaches.

### Findings in relation to previous studies

The characteristics of women in our cohort are broadly similar to all women who give birth in terms of age and mode of delivery in a comparison year of 2017.[28 29] However, our cohort has a greater proportion of women who live in less deprived areas compared with all births in England and Wales, which limits the generalisability of our findings. We identified that those from more

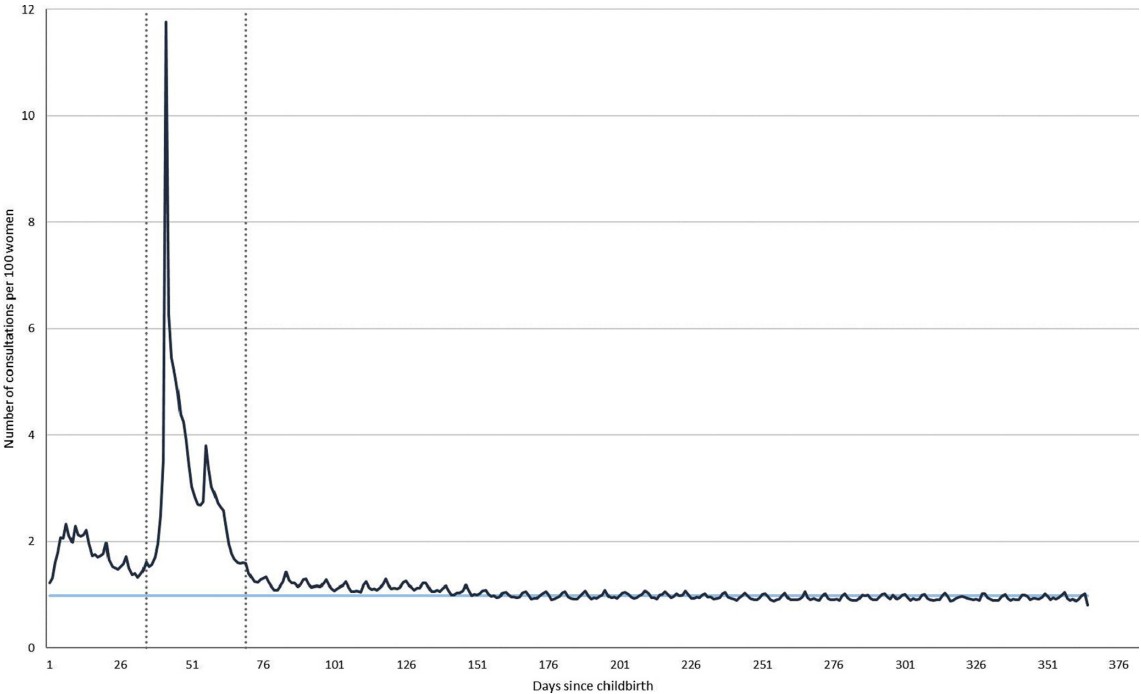

**Figure 2** Women's consultation rate on each day in the first year following childbirth.

deprived areas were less likely to have a postnatal check, which supports previous findings.[20] This may mean we overestimate the proportion of women who have a postnatal check compared with all women who give birth in the UK. Previous studies estimate that 85%–91% of women in England have a postnatal check[18 19]; however only 56% of women in our study had evidence of having one. We would expect our estimates to be lower as we used electronic health records which capture a broader picture of real-world practice compared with previous studies that may have been subject to selection and recall bias. When using a more sensitive approach, where we considered any consultation in weeks 5–10 as evidence of a postnatal check, our findings are closer to those of previous studies, although we are aware that not all of these consultations would have covered topics meant for the postnatal check. It is likely the true number having a check lies between our two estimates. The consultation rate of 4.8 per person per year identified in our study is comparable to that found by others, when taking age, sex and reason for consultation into account.[30]

There are several possible explanations for our findings of a low uptake of postnatal checks. It is possible that women do not want or feel they need advice from GPs; or invitations from the GP are not taken up either because women do not respond to them, or may find it difficult to access appointments. Alternatively a lack of recording in electronic health records may explain the apparently low rate of postnatal checks.

### Implications of findings and future research recommendations

It is encouraging to find that the majority of women return to primary care at least once in the year after childbirth; however, it is concerning that four in 10 women did not have a structured postnatal check documented and that consultation rates have declined over time. In the UK, approximately 800 000 women give birth each year[2]; our estimates (44%) then suggest that up to 350 400 women may be missing this key check. The postnatal period is a potentially vulnerable time for women and there could be serious consequences to not identifying women at risk of poor health or harm after childbirth.[31] The postnatal check has been shown to be a key contact to identify serious health needs such as postnatal depression, which affects one in six women after childbirth.[25] It also provides protected time and opportunities to improve women's health and well-being through preventative intervention, such as timely access to contraception, advice about weight management or diet following gestational weight gain, or support to stop smoking can be given.[4] Our finding that younger women and those from more deprived areas are less likely to have a check is particularly important as they may be most likely to benefit. For example, contraceptive uptake is particularly low in younger and more deprived groups,[32] and offering timely access through the check could lead to fewer unwanted or repeat pregnancies for these women.

Our findings suggest practices may need to implement systems for follow-up of women who have declined or missed a postnatal check. There is a need for better promotion of the benefits of attending the postnatal check at other times in the maternity pathway; such as during midwife or health visitor appointments, in hospital or birth units, or at other GP maternity and baby check-ups. Additionally, there are currently no known financial or quality-based incentives to document primary care activity in the postnatal period. This could lead to variation in services

and underreporting of activity. It is vital to improve the documentation of this care to more accurately understand women's use of the postnatal check, more broadly their health needs and service use after childbirth, and ultimately improve care. This is particularly important as we identified that postnatal consultation rates declined over time. We recommend more research to explore the reasons behind the low uptake of postnatal checks and variation in consultation rates.

In this study our focus was on who had a postnatal check, and while NICE outlines the content of these appointments, few studies have explored what health needs are covered in actuality. This should be explored through further research to better understand what content and delivery are most effective for women and if attending a postnatal check leads to better outcomes. Current NICE guidance recommends the postnatal check take place 6 weeks after childbirth within primary care. There has been some evaluation of this timing and frequency,[33 34] and in particular how this relates to the early postnatal care women receive from midwives. Further high-quality studies are needed to determine the most effective timing of postnatal care consultations. It is also important to examine the accuracy of postnatal care in electronic health records and explore ways to improve this in future studies. Lastly, our study focused on the postnatal care use of women who had given birth to a single live infant only. Complex births, such as multiple deliveries or stillbirths are relatively rare (15.9 out of every 1000 women giving birth in England and Wales had a multiple birth in 2016 and 4.4 per 1000 births were a stillbirth).[35] It is expected that these women would receive additional follow-up in specialist care and so would not represent the usual pathway back to primary care services. However, this has not been well investigated and future studies should explore if these women have different experiences of postnatal care. Furthermore, future studies could explore the differences in postnatal care use by ethnicity, country of birth, language spoken and refugee or asylum seeker status.

## CONCLUSION

Two in 10 women had no consultation at the time of the postnatal check and four in 10 women have no record of receiving a structured postnatal check within the first 10 weeks after giving birth, this is despite the majority of women returning to primary care at least once in the year after childbirth. Teenagers and those from the most deprived areas are among the least likely to have a check. We estimate up to 350 400 women per year in the UK may be missing these opportunities for timely health promotion and to have important health needs identified after childbirth.

**Acknowledgements** We are very grateful to members of the public who generously gave their time to feedback on the outline of this study, in particular we are grateful to the many women who have shared their experiences of childbirth and beyond, and who serve as a continued source of inspiration for this research. We are also appreciative for the support of our funders.

**Contributors** This study was designed and conceived by HCS, SS and IP. HCS conducted the data analysis and wrote the first draft of this manuscript. SS and IP made comments on the manuscript. HCS and IP had full access to the database and guarantee this manuscript is an honest, accurate, and transparent account of the study being reported; that no important aspects of the study have been omitted; and that any discrepancies from the study as planned have been explained.

**Funding** This study is funded by the National Institute for Health Research (NIHR) School for Primary Care Research (SPCR) (project reference 549832). The views expressed are those of the authors and not necessarily those of the NIHR or the Department of Health and Social Care. SS is funded by the National Institute for Health Research (NIHR) School for Public Health Research (SPHR) and NIHR Northwest London Applied Research Collaboration (ARC). The School for Public Health Imperial College London is also grateful for support from the Imperial NIHR Biomedical Research Centre.

**Competing interests** All authors have completed the ICMJE uniform disclosure form at www.icmje.org/coi_disclosure.pdf and declare: HCS reports grants from National Institute of Health Research School for Primary Care Research and SS reports grants from NIHR School for Public Health Research, grants from NIHR NW London Applied Research Collaboration, during the conduct of the study; grants from The Daily Mile Foundation Charity, outside the submitted work

**Patient and public involvement** Patients and/or the public were involved in the design, or conduct, or reporting, or dissemination plans of this research. Refer to the Methods section for further details.

**Patient consent for publication** Not required.

**Ethics approval** Approval was received from the Scientific Review Committee on 10 April 2019 (THIN protocol number: 19THIN013). THIN is a registered trademark of Cegedim SA in the UK and other countries. Reference made to THIN database is intended to be descriptive of the data asset licensed by IQVIA. This work uses de-identified data provided by patients as a part of their routine primary care.

**Provenance and peer review** Not commissioned; externally peer reviewed.

**Data availability statement** No additional data are available as this work draws on de-identified data provided by patients as a part of their routine primary care.

**ORCID iDs**
Holly Christina Smith http://orcid.org/0000-0001-6805-680X
Sonia Saxena http://orcid.org/0000-0003-3787-2083
Irene Petersen http://orcid.org/0000-0002-0037-7524

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
