## [Reviewer comments · BMJ Open]

ARTICLE DETAILS

TITLE (PROVISIONAL)	Postnatal checks and primary care consultations in the year following childbirth: an observational cohort study of 309,573 women in the UK, 2006-2016
AUTHORS	Smith, Holly; Saxena, Sonia; Petersen, Irene

VERSION 1 – REVIEW

REVIEWER	Stephanie Brown Murdoch Children's Research Institute Australia
REVIEW RETURNED	16-Feb-2020

GENERAL COMMENTS	This paper draws on routinely collected drawn from UK primary care electronic medical records to investigate postnatal checks and primary care in the year after childbirth for >300,000 women between 2006 and 2016. This is a complex undertaking, and the authors provide a very clear account of some of the challenges associated with utilising data of this kind. Overall, the paper is well written, and tables and supplementary information provide useful information not generally available to inform policy makers/primary care practitioners reviewing the extent to which current practice matches or varies from NICE guidance. With this in mind, it is important that such information is analysed and interpreted with maximum attention to factors which may enhance or limit inferences. To this end, I have a number of suggestions for the authors' consideration: • In several places in the manuscript (including the summary of strengths and limitations on page 3), the authors describe the cohort used for analysis as both population-based and representative. However, limited comparisons are provided to establish that the latter is true. I note that data are drawn from 730 primary care practices 'across the UK' (p 5), but no data are provided on the geographic location of these practices, the extent to which they are inclusive of urban and rural settings, and different jurisdictions and sub-populations within the UK. More complex births, multiple births and stillbirths are excluded, which limits generalisability for these not uncommon experiences (if considered as a group). It is unclear why they are excluded, since postnatal care remains important, if not, more important for women in these circumstances. In addition, it appears that a sizeable group of women are excluded from analysis due to incomplete data (p5). Given these exclusions, how confident can we be about cohort representativeness, with regard to key covariates, such as maternal age, place of residence, maternal country of birth and fluency in English, mode of birth, and pregnancy complications/complex medical conditions? Each of these factors are important considerations in relation to appropriate tailoring of
---

postnatal primary care, and while data may not be available for all covariates of interest, it would be useful for the authors to compare the cohort characteristics with data for all births in the UK during the study period where possible (not just births recorded in THIN).

- I was pleased to see the inclusion of a supplementary analysis reporting on all postnatal visits between 5-10 weeks, in addition to the data presented on visits identified as a postnatal check in Read/AHD codes recognising the variation in use of these codes by GP practices (p10). Given the widely different figures produced by these 2 approaches to classification (and the noted caveats about the accuracy of coding), why give primacy to the data drawn from using the Read/AHD codes when interpreting and discussing results (p 11). It seems somewhat disingenuous to argue that “four in ten women did not have a postnatal check’ (page 11) given the data presented arguably tell a different story? And further to this, is it truly accurate to state that ‘350,000 women may be missing out on this check’ when the representativeness of the cohort has not been established, and some higher risk groups (ie women with a multiple birth, complex history, or stillbirth) are excluded?

In addition, noted below are some other issues that the authors may wish to consider if revising the manuscript:

- Introduction, page 4, line 18 – duplication of words
- Aims, page 4 – the paper goes beyond the stated aim of reporting on prevalence to explore associated factors, and change over time. It would be useful to articulate these aspects of analyses presented more clearly in the aims (and the abstract).
- Data source, page 5 – are data available on maternal country of birth, language spoken, refugee or asylum seeker status?
- Study population, page 5, line 45 – suggest reword as “ given birth to a single live infant’
- Consultations, page 6 – why were consultations by phone included?
- Patient and childbirth characteristics, page 6 – the use of data on children registered within the same household to determine parity is highly problematic given the number of other reasons why other children may be present in the household (eg. step siblings, extended family households). This needs to be noted as a limitation, and I would strongly encourage the authors not to use the word ‘parity’ as the label for this variable given the way in which it was derived.
- Given the large sample size, was consideration given to reporting results with 99% confidence intervals?
- What proportion of women not recorded (i.e. coded) as having a postnatal check had had a visit in the period preceding this? (ie first 4 weeks postpartum) And for purposes of comparison, what proportion of those not in contact with primary care between 5-10 weeks postpartum, had had a visit in the first 4 weeks? While these early visits are likely to be for specific indications, it is possible some women who have already attended their GP during this period, may not see a need for a further postnatal visit and/or may delay this visit until after 10 weeks either of their own accord, or at their GP’s suggestion. It would be useful to explore the data further to elucidate the extent to which different scenarios may be the case.
- Table 2 Adjusted estimates and Table 3 Crude Consultation Rate – I was surprised that more attention was not paid to the data

	presented by year which suggest declining consultations over time. It would be useful for the authors to draw greater attention to this in reporting results and in discussion.  • Table headings: Suggest revise these to ensure that they are 'stand alone', ie. reader can identify what is reported in the table without reading text of paper. Finally, I think the paper would benefit from further and more detailed review of the extant literature on the postnatal check and efforts to improve women's access to early postnatal care. There are several important trials in this area that are not identified (e.g. MacArthur C et al, Lancet, 2002) and (Gunn J et al, BJOG, 1998), and significant debates regarding content and timing of postnatal care that provide important context for interpreting the data presented.
--	--

REVIEWER	Abiodun Adanikin Queen Mary University of London, UK
REVIEW RETURNED	02-Jun-2020

GENERAL COMMENTS	Thank you for requesting my review of this manuscript which touched on an important maternal health subject – postnatal care. I also want to commend the efforts of the authors. The article reads well, and the purpose is clear. My major comment revolves around the decision to use Read codes/AHD codes as the main estimation and the preferred result for the manuscript, whilst the more sensitive approach of 'any postnatal consultation' was relegated as supplementary findings. I feel a swap would have been better and more accurate for scientific readers and the public in general. Often, people would read only the abstract detailing main findings, which makes the less accurate estimation highly likely to be referenced. Secondly, since current study focuses solely on postnatal care attendance and not the content of care, using the more sensitive method, 'any postnatal consultation' is better and more accurate. It will be difficult to justify accuracy of Read codes/AHD codes when the women could have attended but it was wrongly coded. I would therefore recommend that the more sensitive approach be used as the main result for the manuscript. Furthermore, it would be important to highlight briefly what the current UK guideline [e.g. NICE/RCOG] stipulates with respect to scheduling postnatal appointment. Then, authors can provide estimates of patients' compliance with clinic appointments as per the guidance. It is good to know how many women attended when it's recommended as most beneficial. Invariably, there would be multiple attendance, but when the first visit happened is vital in comparison to current recommendation. In addition, it is good to provide information on attendance rate based on the initial scheduled postnatal appointment following delivery. This would provide an estimate on compliance with first postnatal appointments, unless the information is not available. It may also be possible to estimate attendance rate based on successful visit as per the first postnatal appointment following delivery. Other comments: "We assigned women's smoking status as 'current smoker' (record of smoking at any time in the year after childbirth), 'past smoker' (record of smoking or being an ex-smoker in the two years prior to childbirth and not a current smoker), 'non- smoker' or 'unknown'. – This is slightly confusing, especially the 'current smoker' and 'past
--

	smoker categories. Under this classification, someone who smoked throughout pregnancy and hasn't smoked in early postnatal period is classified 'past smoker' but, the smoking is still recent and has tendency to impact on postnatal health. There is likely going to be dilution of the real effect of smoking during pregnancy (which is still supposedly recent) on postnatal health. A classification which considers: 'smoke before pregnancy', 'smoke during pregnancy and/or postnatal period', and 'non-smoker' would be suggested. Pg. 4 L18 – correct typo "...to evaluate their physical and mental the health mental" pg. 4 L41-44 – Quite unsure what this statement is conveying: "Current research may underestimate maternal morbidity as many women do not report symptoms or may be reluctant to seek support after childbirth". The word 'current' tend to refer to your research, may be the sentence should be re-written. Pg.4 L49-53 – "We found no contemporary studies showing patterns of preventive and responsive postnatal care for women. The aim of this observational cohort study was to examine the prevalence of postnatal checks and primary care consultations for women in the first year after childbirth". There seems to be a disconnect between the two sentences, and it will be good if authors can provide a link between the sentences and more clarity. For instance, that the current observational study seeks at first to determine the prevalence of postnatal care attendance... I guess a part of the research project will still consider the content of postnatal care, which is what the initial statement alludes to. Pg. 5 L37-41 – "THIN is broadly representative of the UK population in terms of demographics, chronic disease and mortality; however, more people live in more affluent areas compared to the general population". – please re-write, it is unclear. Is it that THIN is less accurate for generalization on affluence/socio-economic status? Future research recommendation should also consider health record accuracy, challenges and strengthening. Thank you.
--	--

VERSION 1 – AUTHOR RESPONSE

Reviewer 1		
3. In several places in the manuscript (including the summary of strengths and limitations on page 3), the authors describe the cohort used for analysis as both population-based and representative. However, limited comparisons are provided to establish that the latter is true. I note that data are drawn from 730 primary care practices 'across the UK' (p 5), but no data are provided on the geographic location of these practices, the extent to which they are inclusive of urban and rural settings, and different jurisdictions and sub-populations within the UK.	Here, we have provided an additional comparison between our cohort and available national birth statistics for key characteristics. These data draw from the Office for National Statistics (ONS) live births in England and Wales in 2017 dataset¹ and NHS digital maternity statistics for England 2017-18² (Table A). We have used data from 2017 to provide a snapshot comparison as deprivation information was not published prior to 2017 so a longitudinal comparison was not possible. This comparison demonstrates that our cohort is broadly reflective of all women who gave birth in terms of age distribution and mode of delivery;	Findings in relation to previous studies, Page 10 Table A

	however, our cohort has a clear under-representation of women from the most deprived areas (Table A). This finding is reflective of a previous study exploring the representativeness of THIN database.³ We have added this reflection and how it may impact on our findings to the discussion: “The characteristics of women in our cohort are broadly similar to all women who give birth in terms of age and mode of delivery in a comparison year 2017^{28,29}. However, our cohort has a lower proportion of women from more deprived areas compared to all births in England and Wales, which limits the generalisability of our findings. We identified that those from more deprived areas were less likely to have a postnatal check which supports previous findings²⁰. This may mean we overestimate the proportion of women who have a postnatal check compared to all women who give birth in the UK.” If the editor wishes, we would be happy to include Table A in supplementary materials.	
4. More complex births, multiple births and stillbirths are excluded, which limits generalisability for these not uncommon experiences (if considered as a group). It is unclear why they are excluded, since postnatal care remains important, if not, more important for women in these circumstances.	It was not possible to include women who had a stillbirth or multiple delivery in this study as these were excluded from the pre-existing cohort of women we drew from. We have now added a sentence to the method section to explain this: “This pre-existing cohort of women excluded more complex births, including multiple deliveries (twins, triplets etc.) and those with a known miscarriage, termination or stillbirth, so it is not possible to include these women in our study.” In this study we aimed to present a picture of usual care for the majority of women. Other data sources demonstrate that both stillbirths and multiple deliveries are relatively rare (15.9 out of every 1,000 women giving birth in England and Wales had a multiple birth in 2016 and 4.4 per 1,000 births were a stillbirth)⁴. It is expected that women who have complex births such as these would receive additional follow-up in specialist care and so would not represent the usual pathway back to primary care services. It would be interesting to explore the	Methods – Study population, page 5

	postnatal care use of those with complex deliveries in future studies and we have added this to our future research ideas in the discussion: “Lastly, our study focused on the postnatal care use of women who had given birth to a single live infant only. Complex births, such as multiple deliveries or stillbirths are relatively rare (15.9 out of every 1,000 women giving birth in England and Wales had a multiple birth in 2016 and 4.4 per 1,000 births were a stillbirth)³³ It is expected that these women would receive additional follow-up in specialist care and so would not represent the usual pathway back to primary care services. However, this has not been well investigated and future studies should explore if these women have different experiences of postnatal care.”	Implications of findings and future research recommendations, page 12
5. In addition, it appears that a sizeable group of women are excluded from analysis due to incomplete data (p5). Given these exclusions, how confident can we be about cohort representativeness, with regard to key covariates, such as maternal age, place of residence, maternal country of birth and fluency in English, mode of birth, and pregnancy complications/complex medical conditions? Each of these factors are important considerations in relation to appropriate tailoring of postnatal primary care, and while data may not be available for all covariates of interest, it would be useful for the authors to compare the cohort characteristics with data for all births in the UK during the study period where possible (not just births recorded in THIN).	We have provided additional information which compares the characteristics of women in our study to national statistics of all women who gave birth where available, please see point 3 for detailed response and changes to the manuscript.	Findings in relation to previous studies, Page 10 Table A
6. I was pleased to see the inclusion of a supplementary analysis reporting on all postnatal visits between 5-10 weeks, in addition to the data presented on visits identified as a postnatal check in Read/AHD codes recognising the variation in use of these codes by GP practices	We consulted extensively with clinicians in developing this study. The postnatal check in primary care is a very different consultation to an ordinary appointment, typically a double appointment with a specific checklist and focus on preventive care. We acknowledge that having a consultation at this time would provide an opportunity for a check	

(p10). Given the widely different figures produced by these 2 approaches to classification (and the noted caveats about the accuracy of coding), why give primacy to the data drawn from using the Read/AHD codes when interpreting and discussing results (p 11). It seems somewhat disingenuous to argue that “four in ten women did not have a postnatal check’ (page 11) given the data presented arguably tell a different story? And further to this, is it truly accurate to state that ‘350,000 women may be missing out on this check’ when the representativeness of the cohort has not been established, and some higher risk groups (ie women with a multiple birth, complex history, or stillbirth) are excluded?	but it is not clear that these women received all the care expected of a planned postnatal check or if they consulted for a specific indication. It is therefore important to consider both a restrictive and broader definition. We have now included the supplementary analysis results within the abstract and made them more prominent in the discussion so that anyone reading the summary/key findings of this paper will have full access to this information. The following changes were made to the abstract: “Of these women, 78.7% (243,516) had a consultation at the time of the postnatal check, but only 56.2% (174,061) had a structured postnatal check documented.” “Two in ten women did not have a consultation at the time of the postnatal and four in ten women have no record of receiving a structured postnatal check within the first ten weeks after giving birth.” And the discussion: “We found that eight in 10 women had a consultation at the time of the postnatal check; however, only half of women had a record of receiving a structured postnatal check which means four in ten women (44%) may have missed out.” “Two in ten women had no consultation at the time of the postnatal check and four in ten women have no record of receiving a structured postnatal check within the first ten weeks after giving birth, this is despite the majority of women returning to primary care at least once in the year after childbirth.”	Abstract – results, page 2 Abstract – conclusions, page 2 Discussion – main findings, page 10
---	---	--

		Discussion – conclusion, page 10
7. Introduction, page 4, line 18 – duplication of words	Thank you, this typo has been corrected.	Introduction, Page 4
8. Aims, page 4 – the paper goes beyond the stated aim of reporting on prevalence to explore associated factors, and change over time. It would be useful to articulate these aspects of analyses presented more clearly in the aims (and the abstract).	We have amended the study aims to better reflect this: “The aim of this observational cohort study was to examine the prevalence of postnatal checks, explore factors associated with having a postnatal check and primary care consultations for women in the first year after childbirth.”	Introduction, Page 4
9. Data source, page 5 – are data available on maternal country of birth, language spoken, refugee or asylum seeker status?	Unfortunately, this information isn’t available in this database. We have included this as a suggestion for further research: “Furthermore, future studies could explore the differences in postnatal care use by ethnicity, country of birth, language spoken and refugee or asylum seeker status.”	Discussion - Implications of findings and future research recommendations, page 12
10. Study population, page 5, line 45 – suggest reword as “ given birth to a single live infant’	This wording has been changed to the suggested. We have also changed to the suggested wording where used throughout the manuscript.	Study population, page 5 Abstract – Participants, page 2 Tables – Table 1, page 18 Tables – Table 2, page 19 Tables – Table 3, page 20

11. Consultations, page 6 – why were consultations by phone included?	As phone consultations include direct contact between a patient and health care professional patients may receive many elements of care they would receive in a face-to-face consultation. We felt excluding phone consultations would underestimate the care women received in the year after childbirth.	
12. Patient and childbirth characteristics, page 6 – the use of data on children registered within the same household to determine parity is highly problematic given the number of other reasons why other children may be present in the household (eg. step siblings, extended family households). This needs to be noted as a limitation, and I would strongly encourage the authors not to use the word ‘parity’ as the label for this variable given the way in which it was derived.	We have changed this variable to be called “number of births” throughout the manuscript.	Throughout
13. Given the large sample size, was consideration given to reporting results with 99% confidence intervals?	We had not considered this and chose the 95% convention. We are careful not to draw strongly on the confidence intervals in our interpretation as we know we have a large dataset.	
14. What proportion of women not recorded (i.e. coded) as having a postnatal check had had a visit in the period preceding this? (ie first 4 weeks postpartum) And for purposes of comparison, what proportion of those not in contact with primary care between 5-10 weeks postpartum, had had a visit in the first 4 weeks? While these early visits are likely to be for specific indications, it is possible some women who have already attended their GP during this period, may not see a need for a further postnatal visit and/or may delay this visit until after 10 weeks either of their own accord, or at their GP’s suggestion. It would be useful to explore the data further to elucidate the extent to which different scenarios may be the case.	Thank you for the suggestion. We did explore the data and found that a relatively small proportion of women in our study 18,723 (6.0%) had a consultation in the first four weeks but did not have a consultation in weeks 5-10 (Table B). This compares to a much greater proportion of women 89,605 (28.9%) who had both an early consultation and one in weeks 5-10. As the reviewer suggested it seems as many attending an early consultation will likely be for specific indications, such as infection or breastfeeding support. We have now included this information in the results and Table B in supplementary material: “A small proportion of women in our study 18,723 (6.0%) had a consultation in the first four weeks but did not have a consultation in weeks 5-10 (see supplementary material). This compares to a much greater proportion of women 89,605 (28.9%) who had both an early consultation and one in weeks 5-10.”	Results – Postnatal check, page 8 Supplementary materials Table B

15. Table 2 Adjusted estimates and Table 3 Crude Consultation Rate – I was surprised that more attention was not paid to the data presented by year which suggest declining consultations over time. It would be useful for the authors to draw greater attention to this in reporting results and in discussion.	We have drawn more attention to this finding in our results section: “Consultation rate decreased over time, from 4.9/ person-year (95% CI: 4.9-4.9) in 2006-2007 to 4.2/ person-year in 2014-2015 (4.2-4.3).” and discussion section: “Consultation rates decreased over time (from 4.9 time per woman per year in 2006-2007 to 4.2 times in 2014-2015)” “It is encouraging to find that the majority of women return to primary care at least once in the year after childbirth; however, it is concerning that four in ten women did not have a structured postnatal check documented and that consultation rates have declined over time.” “This is particularly important as we identified that postnatal consultation rates declined over time. We recommend more research to explore the reasons behind the low uptake of postnatal checks and variation in consultation rates.”	Results, Primary care consultations, page 9 Discussion – main findings, page 10 Discussion - Implications of findings and future research recommendations, page 11 Discussion - Implications of findings and future research recommendations, page 12
16. Table headings: Suggest revise these to ensure that they are ‘stand alone’, ie. reader can identify what is reported in the table without reading text of paper.	We have added further details to these so that they have enough information to stand alone. These have been changed to: “Table 1: Characteristics of women who have given birth to a single live infant and proportion of those women with a postnatal check 5-10 weeks after childbirth” “Table 2: Mixed-effects Poisson estimates of the likelihood of having a postnatal check for women who had given birth to a single live infant by age, Townsend score, mode of delivery, number of births, smoking status and year group; unadjusted, and	Tables, pages 17-19

	adjusted for age and deprivation” “Table 3: Crude consultation rate in the first year after childbirth of women who had given birth to a single live infant per/ person-year, by characteristic”	
17. Finally, I think the paper would benefit from further and more detailed review of the extant literature on the postnatal check and efforts to improve women’s access to early postnatal care. There are several important trials in this area that are not identified (e.g. MacArthur C et al, Lancet, 2002) and (Gunn J et al, BJOG, 1998), and significant debates regarding content and timing of postnatal care that provide important context for interpreting the data presented.	We thank the reviewer for their suggestions. To expand on this point, we have we have provided additional context and recommendations in the discussion: “Current NICE guidance recommends the postnatal check take place 6 weeks after childbirth within primary care. There has been some evaluation of this timing and frequency,^{33,34} and in particular how this relates to the early postnatal care women receive from midwives. Further high-quality studies are needed to determine the most effective timing of postnatal care consultations.”	Discussion - Implications of findings and future research recommendations, page 11
Reviewer 2		
18. My major comment revolves around the decision to use Read codes/AHD codes as the main estimation and the preferred result for the manuscript, whilst the more sensitive approach of ‘any postnatal consultation’ was relegated as supplementary findings. I feel a swap would have been better and more accurate for scientific readers and the public in general. Often, people would read only the abstract detailing main findings, which makes the less accurate estimation highly likely to be referenced. Secondly, since current study focuses solely on postnatal care attendance and not the content of care, using the more sensitive method, ‘any postnatal consultation’ is better and more accurate. It will be difficult to justify accuracy of Read codes/AHD codes when the women could have attended but it was wrongly coded. I would therefore recommend that the more sensitive approach be used as the main result for the manuscript.	Many thanks for the suggestion, please see our response to point 6 above. We have made following changes to the manuscript as a result. In the abstract: “Of these women, 78.7% (243,516) had a consultation at the time of the postnatal check, but only 56.2% (174,061) had a structured postnatal check documented.” “Two in ten women did not have a consultation at the time of the postnatal and four in ten women have no record of receiving a structured postnatal check within the first ten weeks after giving birth.” And the discussion: “We found that eight in 10 women had a consultation at the time of the postnatal check; however, only half of women had a record of receiving a structured postnatal check which means four in ten women	Abstract – results, page 2 Abstract – conclusions, page 2

	(44%) may have missed out.” “Two in ten women had no consultation at the time of the postnatal check and four in ten women have no record of receiving a structured postnatal check within the first ten weeks after giving birth, this is despite the majority of women returning to primary care at least once in the year after childbirth.”	Discussion – main findings, page 10 Discussion – conclusion, page 10
19. Furthermore, it would be important to highlight briefly what the current UK guideline [e.g. NICE/RCOG] stipulates with respect to scheduling postnatal appointment. Then, authors can provide estimates of patients’ compliance with clinic appointments as per the guidance. It is good to know how many women attended when it’s recommended as most beneficial. Invariably, there would be multiple attendance, but when the first visit happened is vital in comparison to current recommendation.	The UK postnatal check is recommended to take place 6 weeks after childbirth. From Figure 2 of our manuscript there is a clear peak in consultations in that week. To allow for real-world variation in scheduling we expanded this window to weeks 5-10. In our study, the outcome was binary which meant a women could only have one record of a postnatal check in this window even if there were multiple entries (although this was rare). We have added more context to the scheduling in the discussion: “Current NICE guidance recommends the postnatal check take place 6 weeks after childbirth within primary care. There has been some evaluation of this timing and frequency,^{33,34} and in particular how this relates to the early postnatal care women receive from midwives. Further high-quality studies are needed to determine the most effective timing of postnatal care consultations.”	Discussion - Implications of findings and future research recommendations, page 11
20. In addition, it is good to provide information on attendance rate based on the initial scheduled postnatal appointment following delivery. This would provide an estimate on compliance with first postnatal appointments, unless the information is not available. It may also be possible to estimate attendance rate based on successful visit as per the first postnatal appointment following delivery.	The first planned/scheduled appointment is the six-week postnatal check. Some women may have an earlier appointment sometimes referred to as a ‘postnatal visit’ but this isn’t recommended for all women, is provided on a case-by-case basis and has no set formal structure. It was not possible to examine those who were invited and didn’t attend their postnatal check as this wasn’t consistently recorded across practices and the procedures may vary.	
21. “We assigned women’s smoking status as ‘current smoker’ (record of smoking at any time in the year after childbirth), ‘past	We felt that capturing those who smoked following pregnancy would also capture those who smoked during pregnancy, as it unlikely someone who	

smoker' (record of smoking or being an ex-smoker in the two years prior to childbirth and not a current smoker), 'non- smoker' or 'unknown'. – This is slightly confusing, especially the 'current smoker' and 'past smoker' categories. Under this classification, someone who smoked throughout pregnancy and hasn't smoked in early postnatal period is classified 'past smoker' but, the smoking is still recent and has tendency to impact on postnatal health. There is likely going to be dilution of the real effect of smoking during pregnancy (which is still supposedly recent) on postnatal health. A classification which considers: 'smoke before pregnancy', 'smoke during pregnancy and/or postnatal period', and 'non-smoker' would be suggested.	smoked during pregnancy would stop immediately after childbirth. We compared our figures to national statistics but did not include this in the manuscript due to space restrictions. As context, in our study the proportion of current smokers is 13.1% (excluding those with unknown status). This is comparable to English data on Women's Smoking Status at Time of Delivery, which for our study period (2006-2007 to 2014-15) is 13.4%⁵. This national figure captures those at time of childbirth which reflects those smoking in pregnancy/ the early postnatal period.	
22. Pg. 4 L18 – correct typo "...to evaluate their physical and mental the health mental"	Thank you, this typo has been corrected.	Introduction, Page 4
23. pg. 4 L41-44 – Quite unsure what this statement is conveying: "Current research may underestimate maternal morbidity as many women do not report symptoms or may be reluctant to seek support after childbirth". The word 'current' tend to refer to your research, may be the sentence should be re-written.	The wording has been changed to the suggested.	Introduction, Page 4
24. Pg.4 L49-53 – "We found no contemporary studies showing patterns of preventive and responsive postnatal care for women. The aim of this observational cohort study was to examine the prevalence of postnatal checks and primary care consultations for women in the first year after childbirth". There seems to be a disconnect between the two sentences, and it will be good if authors can provide a link between the sentences and more clarity. For instance, that the current observational study seeks at first to determine the prevalence of postnatal care attendance... I guess a part of the research project will still consider the content of postnatal care, which is what the initial statement alludes to.	Thanks for the suggestion, sentences have been reworded to be clearer: "We found no contemporary studies showing patterns of primary care use for women following childbirth. The aim of this observational cohort study was to examine the prevalence of postnatal checks, explore factors associated with having a postnatal check and primary care consultations for women in the first year after childbirth."	Introduction, Page 4
25. Pg. 5 L37-41 – "THIN is broadly representative of the UK population in	The sentence has been reworded to be clearer:	Data source, pg5

terms of demographics, chronic disease and mortality; however, more people live in more affluent areas compared to the general population". – please re-write, it is unclear. Is it that THIN is less accurate for generalization on affluence/socio-economic status?	"However, there is an over-representation of more affluent people as there is a greater proportion of people who live in less deprived areas contributing to THIN database compared to the UK general population²⁴."	
26. Future research recommendation should also consider health record accuracy, challenges and strengthening.	We have added this suggestion to the discussion: "It is also important to examine the accuracy of postnatal care in electronic health records and explore ways to improve this in future studies."	Implications of findings and future research recommendations, page 12

Additional information

Table A: Baseline comparing characteristics of study cohort to national statistics of all women who gave birth in 2017

Characteristic	Study cohort N, %	National statistics N, %	
Overall	309,573	679,106 ¹	
Maternal age (years)		Maternal age (years)¹	
15-19	9,568 (3.1)	<20	20,358 (3.0)
20-24	43,116 (13.9)	20-24	97,506 (14.4)
25-29	77,698 (25.1)	25-29	190,028 (28.0)
30-34	98,269 (31.7)	30-34	216,787 (31.9)
35-39	64,171 (20.7)	35-39	125,114 (18.4)
40-44	15,908 (5.1)	40-44	26,956 (4.0)
45-49	843 (0.3)	45-49	2,357 (0.3)
Townsend Score quintile		Index of Multiple Deprivation (IMD)¹	
1-least deprived	58,583 (21.1)	10 – least deprived	46,907 (6.9)
2	53,656 (19.4)	9	53,341 (7.9)
3	62,023 (22.4)	8	56,573 (8.3)
4	58,506 (21.1)	7	58,381 (8.6)
5-most deprived	44,346 (16.0)	6	62,819 (9.3)
Missing	32,459	5	65,758 (9.7)
		4	72,895 (10.7)
		3	80,647 (11.9)
		2	86,529 (12.7)
		1 – most deprived	95,120(14.0)
		Unknown	136
Overall	309,573	626,203 ²	
Mode of delivery		Mode of delivery²	
Vaginal delivery	75,506 (76.3)	Vaginal delivery	438,924 (71.2)
Caesarean	23,426 (23.7)	Caesarean	177,793 (28.8)

Unknown	210,641	Unknown	9,486
---------	---------	---------	-------

1 – data for maternal age and deprivation based on Office for National Statistics (ONS) live birth statistics for England and Wales in 2017¹. 2 – data for mode of delivery based on NHS digital maternity statistics for England 2017-18². Proportions exclude unknown or missing categories.

Table B: Number of women who had a consultation in weeks 0-4 and/or weeks 5-10, % of cohort

Consultation in week 0-4 (down)/ Consultation in weeks 5-10 (across)	Yes	No	Total
Yes	89,605 (28.9%)	18,723 (6.0%)	108,328 (35.0%)
No	153,911 (49.7%)	47,334 (15.3%)	201,245 (65.0%)
Total	243,516 (78.7%)	66,057 (21.3%)	309,573

References

- Office for National Statistics. Births by Parents' Characteristics, England and Wales [data file]. <https://www.ons.gov.uk/peoplepopulationandcommunity/birthsdeathsandmarriages/livebirths/datasets/birthsbyparentscharacteristics>. Published 2017. Accessed June 18, 2020.
- NHS Digital. NHS Maternity Statistics, 2017-18: Hospital Episode Statistics [data file]. <http://digital.nhs.uk/pubs/maternity1718>. Published 2018. Accessed June 18, 2020.
- Blak BT, Thompson M, Dattani H, Bourke A. Generalisability of the Health Improvement Network (THIN) database: Demographics, chronic disease prevalence and mortality rates. *Inform Prim Care*. 2011;19(4):251-255. doi:10.14236/jhi.v19i4.820
- Office for National Statistics. Birth Characteristics in England and Wales [data file]. <https://www.ons.gov.uk/peoplepopulationandcommunity/birthsdeathsandmarriages/livebirths/datasets/birthcharacteristicsinenglandandwales>. Published 2016. Accessed June 18, 2020.
- NHS Digital. Provisional statistics on Women's Smoking Status at Time of Delivery: England, Quarter 3, 2019-20 [data file].

VERSION 2 – REVIEW

REVIEWER	Abiodun Adanikin Queen Mary University of London, United Kingdom
REVIEW RETURNED	26-Jul-2020

GENERAL COMMENTS	Thank you for the effort at revising the manuscript. It is much improved and reads well. I have few minor comments: Page 5 L37-44: "THIN is broadly representative of the UK population in terms of demographics, chronic disease and mortality. However, there is an over-representation of more affluent people as there is a greater proportion of people who live in less deprived areas contributing to THIN database compared to the UK general population". – I still find this difficult to understand. There is 'over-
---

	representation' of more affluent people in the database, at this same time, there is GREATER proportion of people who live in less deprived areas contributing to the database. It is either there is fair balance in socioeconomic representation, or a group is over-represented. Why not simply say, "However, there is an over-representation of more affluent people (or less deprived people)", whichever is the case and the sentence can stop there. Page 9, L30-32: "Overall, just over half of the women in our study (56%) had a postnatal check"- add 'formal/structured' postnatal check Page 10 L12-14: delete "which means four in ten women (44%) may have missed out". Page 10 L 59 to Page 11 L1-2: "However, our cohort has a LOWER proportion of women from more deprived areas..." – earlier, Page 5 L37-44 indicates 'there is a GREATER proportion of people who live in less deprived areas contributing to THIN database'. This seems conflicting, having lower and greater proportion of the same sub-population contributing to database/analysis. Since this impact on generalizability, it is good that this is properly clarified for readers. Then, the statements: "We also identified that those from more deprived areas were less likely to have a postnatal check which supports previous findings²⁰. This may mean we overestimate the proportion of women who have a postnatal check compared to all women who give birth in the UK" - can be correctly interpreted. There are few typos/omissions that could be addressed during copy-editing.
--	---

VERSION 2 – AUTHOR RESPONSE

Reviewer 2		
2. Page 5 L37-44: "THIN is broadly representative of the UK population in terms of demographics, chronic disease and mortality. However, there is an over-representation of more affluent people as there is a greater proportion of people who live in less deprived areas contributing to THIN database compared to the UK general population". – I still find this difficult to understand. There is 'over-representation' of more affluent people in the database, at this same time, there is GREATER proportion of people who live in less deprived areas contributing to the database. It is either there is fair balance in socioeconomic representation, or a group is over-represented. Why not simply say, "However, there is an over-representation of more affluent people (or less deprived people)", whichever is the case and the sentence can stop there.	We have changed the sentence to the suggested wording: "THIN is broadly representative of the UK population in terms of demographics, chronic disease and mortality; however, there is an over-representation of more affluent people ²⁴."	Data source, Page 5

3. Page 9, L30-32: “Overall, just over half of the women in our study (56%) had a postnatal check”- add ‘formal/structured’ postnatal check	We have changed the sentence to the suggested wording: “Overall, just over half of the women in our study (56%) had a structured postnatal check, i.e. 44% had no such records (Table 1).”	Results – Postnatal check, page 8
4. Page 10 L12-14: delete “which means four in ten women (44%) may have missed out”.	We have removed this extra wording as suggested. This sentence now reads: “We found that eight in 10 women had a consultation at the time of the postnatal check; however, only half of women had a record of receiving a structured postnatal check.”	Discussion – Main findings, page 10
5. Page 10 L 59 to Page 11 L1-2: “However, our cohort has a LOWER proportion of women from more deprived areas...” – earlier, Page 5 L37-44 indicates ‘there is a GREATER proportion of people who live in less deprived areas contributing to THIN database’. This seems conflicting, having lower and greater proportion of the same sub-population contributing to database/analysis. Since this impact on generalizability, it is good that this is properly clarified for readers. Then, the statements: “We also identified that those from more deprived areas were less likely to have a postnatal check which supports previous findings²⁰. This may mean we overestimate the proportion of women who have a postnatal check compared to all women who give birth in the UK” - can be correctly interpreted.	We have changed the sentence to the suggested wording: “However, our cohort has a greater proportion of women who live in less deprived areas compared to all births in England and Wales, which limits the generalisability of our findings.”	Discussion – Study strengths and weaknesses, page 10